# Macromolecular sheets direct the morphology and orientation of plate-like biogenic guanine crystals

Avital Wagner [1], Alexander Upcher [2], Raquel Maria [2], Thorolf Magnesen[3], Einat Zelinger[4], Graça Raposo [5,6] & Benjamin A. Palmer [1] ✉

Animals precisely control the morphology and assembly of guanine crystals to produce diverse optical phenomena in coloration and vision. However, little is known about how organisms regulate crystallization to produce optically useful morphologies which express highly reflective crystal faces. Guanine crystals form inside iridosome vesicles within chromatophore cells called iridophores. By following iridosome formation in developing scallop eyes, we show that pre-assembled, fibrillar sheets provide an interface for nucleation and direct the orientation of the guanine crystals. The macromolecular sheets cap the (100) faces of immature guanine crystals, inhibiting growth along the π-stacking growth direction. Crystal growth then occurs preferentially along the sheets to generate highly reflective plates. Despite their different physical properties, the morphogenesis of iridosomes bears a striking resemblance to melanosome morphogenesis in vertebrates, where amyloid sheets template melanin deposition. The common control mechanisms for melanin and guanine formation inspire new approaches for manipulating the morphologies and properties of molecular materials.

Understanding how organisms control the formation of biogenic crystals is the central, widely studied question of (inorganic) biomineralization[1–6]. However, the mechanisms underlying the formation of organic bio-crystals, which are ubiquitous in animals[6–9] and plants[10], remain a mystery. Guanine crystals are the most widespread molecular bio-crystal. The crystals are constructed from π-stacked, H-bonded layers[11], and exhibit exceptional optical properties, due to the extreme in-plane refractive index ($n = 1.83$) (Supplementary Fig. 1). By precisely controlling the shape and assembly of these crystals, organisms produce an array of different optical phenomena used in camouflage[12], display[13], and vision[14,15]. From a single material, an impressive variety of crystal morphologies are generated, including regular squares and hexagons, irregular polygons, and prisms[16]. All these crystals share a common feature—they preferentially express the highly reflective (100) face (parallel to the high-index, H-bonded layers) to maximize reflection. In contrast, guanine crystals grown in vitro from aqueous solutions grow preferentially along the π-stacking direction resulting in the expression of low-index (-1.45) crystal faces. It is not known how organisms regulate crystallization to generate optically useful morphologies, but recent studies appear to have ruled out several hypotheses: (i) the use of purines as intracrystalline growth additives[16], (ii) molding of amorphous precursor phases[17], and (iii) physical shaping by the crystal organelle membrane[17].

The little we know about guanine formation is that the crystals form inside membrane-bound vesicles called iridosomes[18,19] within specialized chromatophore cells called iridophores. In vertebrates,

[1]Department of Chemistry, Ben-Gurion University of the Negev, Beer-Sheba 8410501, Israel. [2]Ilse Katz Institute for Nanoscale Science & Technology, Ben-Gurion University of the Negev, Beer-Sheba 8410501, Israel. [3]Department of Biological Sciences, University of Bergen, Postbox 7803 Bergen N-5020, Norway. [4]The CSI Center for Scientific Imaging, The Robert H. Smith Faculty of Agriculture, Food and Environment, The Hebrew University of Jerusalem, POB 12 Rehovot 7610001, Israel. [5]Institut Curie, PSL Research University, CNRS, UMR144, Structure and Membrane Compartments, 75005 Paris, France. [6]Institut Curie, PSL Research University, CNRS, UMR144, Cell and Tissue Imaging Facility (PICT-IBiSA), 75005 Paris, France. ✉e-mail: bpalmer@bgu.ac.il

iridophores differentiate from the same progenitor cell as melanophores (melanin pigment cells) and xanthophores (pteridine/carotenoid pigment cells)[20]. Bagnara et al. postulated that all vertebrate pigment organelles ("-somes") derive from a common "primordial vesicle"[21], originating from the endosomal pathway[19]. Furthermore, it was shown that many of these pigment granules are lysosome-related organelles (LRO)[22–24]. Despite their inclusion in the LRO family, iridosomes differ from other pigments in that they generate structural colors from crystals, rather than absorptive colors from pigments. In contrast to other pigment organelles[19], iridosome biogenesis has not been explored.

By following the formation of guanine crystals in juvenile scallop eyes, we show that, in iridosomes, pre-assembled macromolecular sheets template the nucleation of guanine on the planar face of the aromatic molecule. The sheets then cap the (100) faces of the immature crystals, inhibiting growth along the π-stacking direction, and directing growth along the orthogonal H-bonding direction to form highly reflective plates. The morphogenesis of iridosomes bears a striking resemblance to melanosome organellogenesis where melanin is deposited on templating amyloid fibrils. This parallel suggests that common control mechanisms underly the formation of these two pigment organelles.

## Results

The image-forming mirror of scallop eyes illustrates the precise control which organisms exert over crystal morphology and assembly. *Pecten maximus* scallops have hundreds of eyes (Fig. 1a), each containing a concave mirror (Fig. 1b, pseudo-colored green)

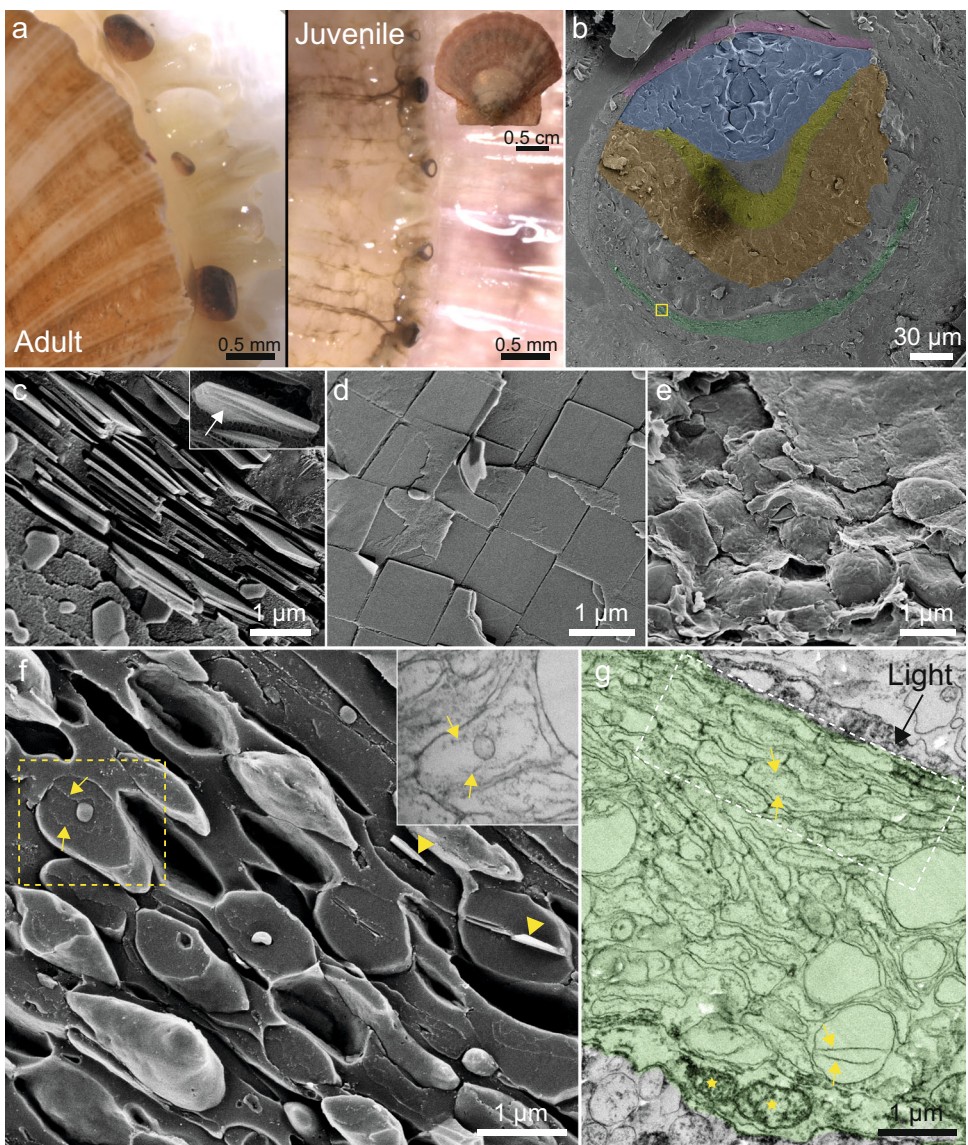

**Fig. 1 | The image-forming mirror in the eyes of an adult and juvenile scallops. a** Stereomicroscope images of numerous eyes in the upper mantle of an adult (left) and a juvenile (right) scallop. Insert; a juvenile scallop (approximately 20 mm in diameter). **b** Pseudo-colored cryo-SEM micrograph of a freeze-fractured eye of a juvenile scallop approximately 230 μm in diameter. Purple; cornea, blue; lens, yellow; distal retina, orange; proximal retina, green; the mirror region. The yellow box is the area shown in high magnification in (**f**). **c, d** Cryo-SEM micrographs of the mirror region in an adult scallop viewed perpendicular to (**c**) and along (**d**) the optic axis. Insert in (**c**) shows the delimiting membrane of the iridosome (white arrow) around a fully formed crystal. **e, f** Cryo-SEM micrographs of the mirror in a juvenile scallop eye viewed along (**e**) and perpendicular to (**f**) the optic axis (the yellow box in (**b**)). The mirror is composed of ellipsoidal iridosomes at various formation stages. Insert in (**f**); SEM image of an iridosome taken with a STEM detector showing an example of a vesicle containing two intraluminal fibrils. **g** SEM image taken with STEM detector of an ultra-thin tissue section, showing the mirror in the juvenile scallop eye pseudo-colored green. In (**f**) and (**g**), a variety of iridosome formation states are observed. **f, g**: yellow arrows; intraluminal fibrils, yellow stars; mitochondria, white dashed box; an area of tessellated vesicles.

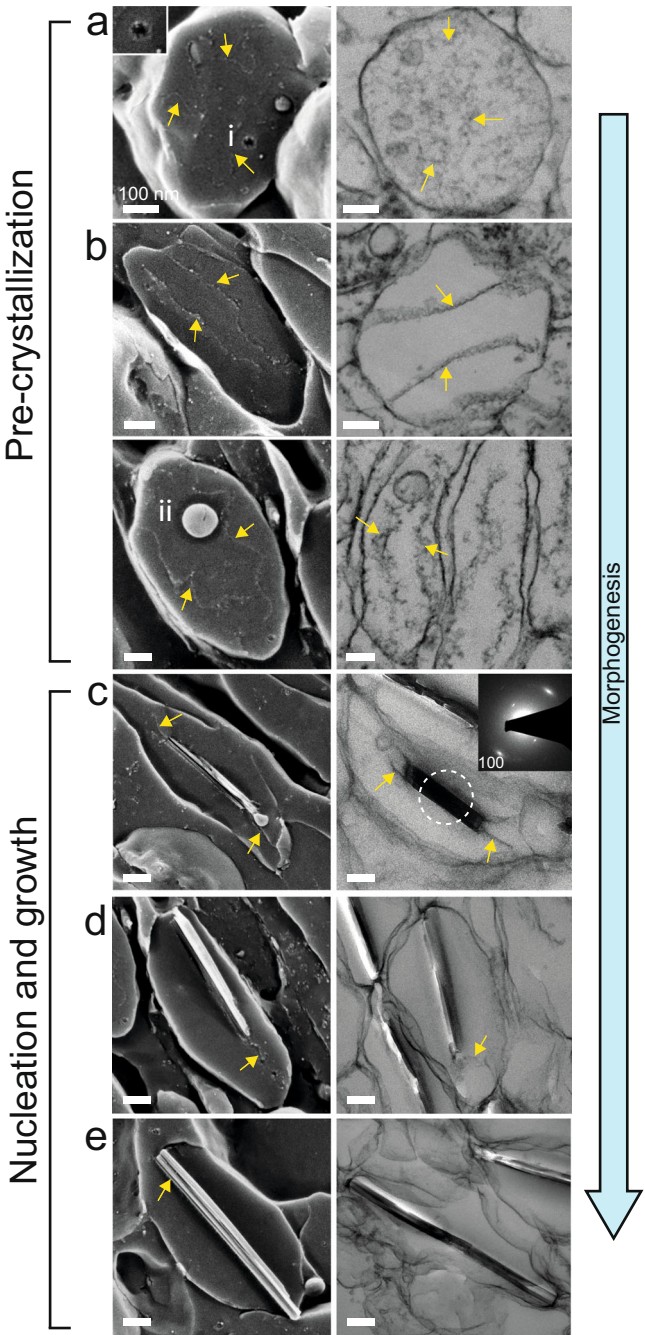

**Pre-crystallization**

**Nucleation and growth**

Morphogenesis

**Fig. 2 | The morphogenesis of guanine crystals in a juvenile scallop eye by cryo-SEM (left column) and TEM (right column). a**, **b** Pre-crystallization: **a** Spherical vesicles containing small type i intraluminal vesicles and some disordered fibrils. Insert; high magnification image of a type i intraluminal vesicle. **b** Ellipsoidal iridosomes are characterized by two intraluminal fibrils stretching across the length of the vesicle. Ellipsoidal iridosomes prior to crystallization are characterized by two types of intraluminal vesicles. Type ii ILVs shown here, are bound to the intraluminal fibers. Type iii ILVs are typically unbound to the fibrils and are shown in Supplementary Fig. 5. **c**–**e** Nucleation and growth: **c** As morphogenesis proceeds, the iridosomes elongate and a small guanine crystal nucleates between the fibrils. The thin immature crystals are elongated along the H-bonded direction and exhibit an electron diffraction pattern characteristic of $\beta$-guanine (insert). **d** The intraluminal fibrils template the growth of the crystal along the H-bonded direction. The fibrils become tightly bound to the (100) face of the crystals. **e** Mature crystals stretch and re-shape the iridosome membrane, which condenses on the surface of the faceted crystal. Yellow arrows; intraluminal fibrils. The juvenile scallops were approximately 2 cm in diameter, with eyes averaging 200 μm in diameter. Scale bars: 100 nm.

squares of the adult (Fig. 1d). Cryo-SEM images of longitudinal sections, show that the mirror is filled with spherical and ellipsoidal iridosomes that, when elongated, have their long axis oriented perpendicular to the incident light (Fig. 1f, g). A variety of morphologically distinct iridosomes are observed in juvenile eyes, representing different stages of iridosome development (Fig. 1f, g). Iridosomes of different developmental stages are often arranged in a gradient from the distal to the proximal sides of the forming iridophore cell (Fig. 1g and Supplementary Fig. 3): Mature iridosomes which are typically found at the distal edge of the cell (towards the incident light direction), are elongated, contain partially formed crystals (Fig. 1f yellow arrowheads) and display the characteristic tessellated suprastructure of adults (Fig. 1e–g). Immature iridosomes, located proximally in the cell, are more spherical and lack crystals (Fig. 1f, g). Prevalent features of these immature iridosomes are two intraluminal fibrils (Fig. 1f, g, yellow arrows) and intraluminal vesicles (ILVs) of various sizes (Fig. 1f, g). The distinct iridosome morphological states, together with proposed morphogenesis, derived by following guanine nucleation and growth, are shown in Fig. 2 (as well as additional examples in Supplementary Fig. 4).

The proximal region of the iridophore cell contains numerous spherical vesicles ~600 nm in diameter, which contain multiple smaller (25–50 nm) intraluminal vesicles (ILVs) and some disordered fibrils (Fig. 2a). The ~600 nm vesicles closely resemble "pre-melanosomes"− multi-vesicular bodies (MVBs) characterizing the first stage of melanosome organellogenesis[27]. We thus assign these vesicles to the most premature iridosomes. The small ILVs (type i ILVs), which have a rough surface texture (Fig. 2a insert and Supplementary Fig. 5), are observed only in spherical iridosomes in the earliest stages of formation.

During the next formation stage, the iridosomes elongate concomitantly with the formation of two intraluminal fibrils, which stretch from one pole of the vesicle to the other (Fig. 2b)[28]. Two types of intraluminal vesicles are seen in these iridosomes, distinguished by their size and degree of association with the intraluminal fibrils. Type ii ILVs (80–100 nm) are closely bound to the organized fibrils, and type iii ILVs (250–325 nm) are typically found near the iridosome membrane and do not associate closely with the fibrils (Supplementary Fig. 5).

As morphogenesis proceeds, an immature guanine crystal, composed of distinct platelets, nucleates in between the two fibrils (Fig. 2c). Often, these crystals appear to be bound to one or both of the two fibrils (Fig. 2 and Supplementary Fig. 4). The crystalline nature of the deposit is confirmed by in situ electron diffraction patterns which exhibit the characteristic signature of $\beta$-guanine. The guanine crystal then grows in the H-bonding direction along the fibrils, which are tightly bound to the (100) crystal faces (Fig. 2d, e). Eventually, the growing crystal contacts both sides of the iridosome, pulling the membrane around the growth front of the crystal. The reshaping of the

which focuses light onto the overlying retinas[25]. The mirror, which is contained in an iridophore cell (Supplementary Fig. 2), is formed from tessellated layers of square guanine crystals (Fig. 1c, d)—a crystal habit forbidden by the monoclinic symmetry of guanine[11,26]. Each crystal is tightly bound by an enveloping membrane (i.e., an iridosome organelle)[25] (insert; Fig. 1c). The highly reflective (100) face of the crystals is oriented towards the incident light across the entire surface of the mirror to direct light to focal points on the two retinas[14,25].

To investigate the formation of the unusual guanine crystals, we examined juvenile (~two months post-hatching) scallop eyes (Fig. 1a, b) using cryogenic scanning (cryo-SEM) and transmission electron microscopy (TEM). The 10–20 mm scallops (Fig. 1a), have dozens of minute eyes, 150 to 250 μm in diameter (Fig. 1b). In juveniles, the mirror is composed of loosely tessellated, "pillow-shaped" iridosomes (Fig. 1e) rather than the perfectly tiled

iridosome vesicle by the growing crystal was also observed in spiders[17] and lizards[29], and again shows that crystal habit is not dictated by physical confinement by the delimiting vesicle membrane.

In adult scallops, the iridosome membrane (Fig. 3 white arrow) becomes attached to the crystal surface, assuming its faceted shape[25]. Upon closer examination, it is possible to decern a second, inner layer fused directly to the crystal surface (Fig. 3 red arrow). We presume that this layer is composed of the intraluminal fibrils found in immature iridosomes that, together with the vesicle membrane, form a two-tiered envelope around the crystal. This finding provides a possible explanation for previous reports of "double-membrane" delimited iridosomes in spiders, fish, amphibians, and reptiles[18] (Supplementary Fig. 6).

To further investigate the nature and function of the intraluminal fibrils, we performed TEM tomography on iridosomes from juvenile scallops. Three-dimensional tomographic reconstructions of early iri-

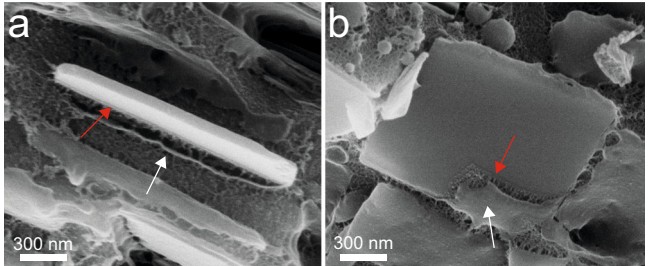

**Fig. 3 | Cryo-SEM images of guanine crystals from an adult scallop eye. a** Side view. **b** Top-down view of guanine crystals showing both the outer lipid vesicle membrane (white arrows) and an inner layer tightly bound to the crystal (red arrows). We assume the inner enveloping layer is the fibrillar sheets observed in juvenile scallop iridosomes which become fused to the crystal surface.

dosomes revealed that the two intraluminal "fibrils" are in fact 2D sheets (Fig. 4a, b and Supplementary Movies 1, 2). These sheets delineate a volume within which guanine crystals form and appear to provide an interface for nucleation (Fig. 2 and Supplementary Fig. 4). Once a crystal is formed, physical capping of the (100) faces by these sheets appears to inhibit growth along the π-stacking axis, directing the crystals to grow along H-bonding direction which results in platelet crystals expressing highly reflective (100) faces. Observations of intercalating sheets in guanine crystals in spiders[17] suggests that templated crystal growth could be a general feature of biogenic guanine formation.

Cryo-SEM micrographs show that the intraluminal sheets are decorated with globular structures (Fig. 4c; white arrows). The immature crystals bound to these sheets are constructed from 10–13 nm thick H-bonded platelets (Fig. 4d, e and Supplementary Fig. 7). In some cases, more than one crystal forms on the sheets (Supplementary Fig. 8). As crystallization proceeds, the platelets coalesce to form thicker (25 nm) crystallites (Fig. 4e). The platelet nature of the crystals may be an intrinsic feature of guanine crystallization occurring independently of the sheets. Similar platelets which merge during crystallization are also observed in spiders[17] and lizards[29], indicating that this is a universal feature of guanine bio-crystallization.

Whilst "capping" provides an answer to the formation of flat crystals, control over crystal twinning is the likely means of generating squares[26]. Scallop crystals are comprised of three monoclinic domains, twinned about the (100) plane. Evidence of this twinning is seen in the platelet texture of the crystals. Hirsch et. al., suggested[26] that by controlling the number of domains and angles between twins, a crystal with a high symmetry morphology can be obtained from a low symmetry structure.

In addition to templating guanine nucleation and growth, the intraluminal sheets play a key role in controlling crystal

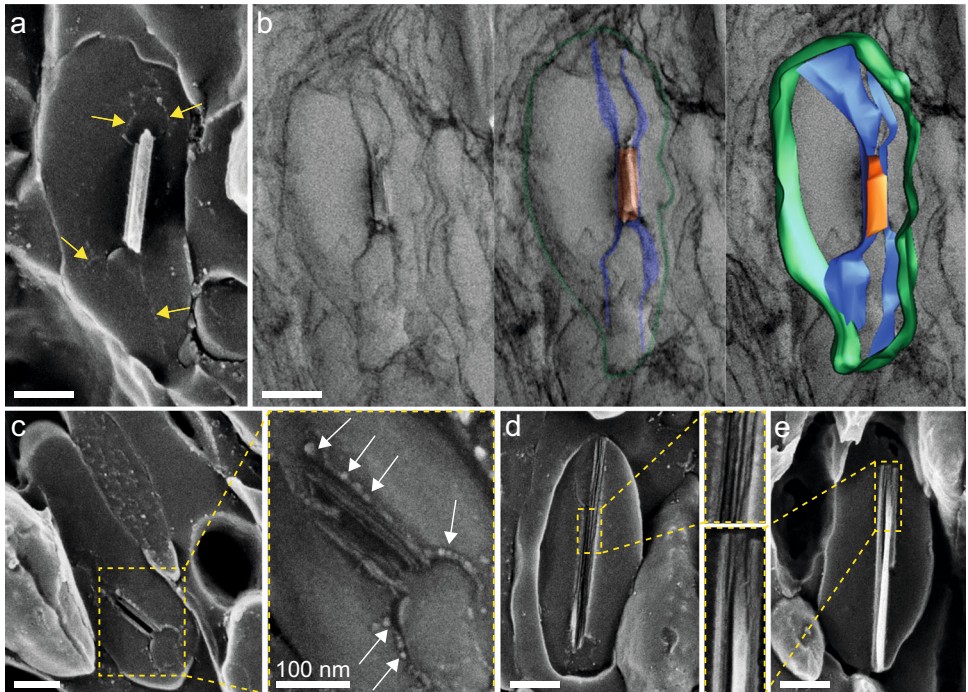

**Fig. 4 | Intraluminal sheets control crystal nucleation and growth.** Electron microscopy images of iridosomes from a juvenile scallop eye 200 μm in diameter. **a** Cryo-SEM micrograph of an immature guanine crystal sandwiched between two intraluminal fibrils (yellow arrows). **b** Left; TEM micrograph of an iridosome containing an immature guanine crystal. Middle; pseudo-colored rendering of the same iridosome (green; iridosome membrane, blue; intraluminal sheets, orange; crystal). Right; image-segmentation obtained from 3D TEM tomography overlaid on the raw 2D TEM image, revealing that the intraluminal fibrils are 2D sheets. **c** Cryo-SEM image of a very immature crystal nucleating on the intraluminal sheets, which are decorated with tiny globules (white arrows). **d** Immature crystals are composed of 10–13 nm thick platelets, which **e** coalesce to form thicker (25 nm) crystallites during maturation. Scale bars: 200 nm.

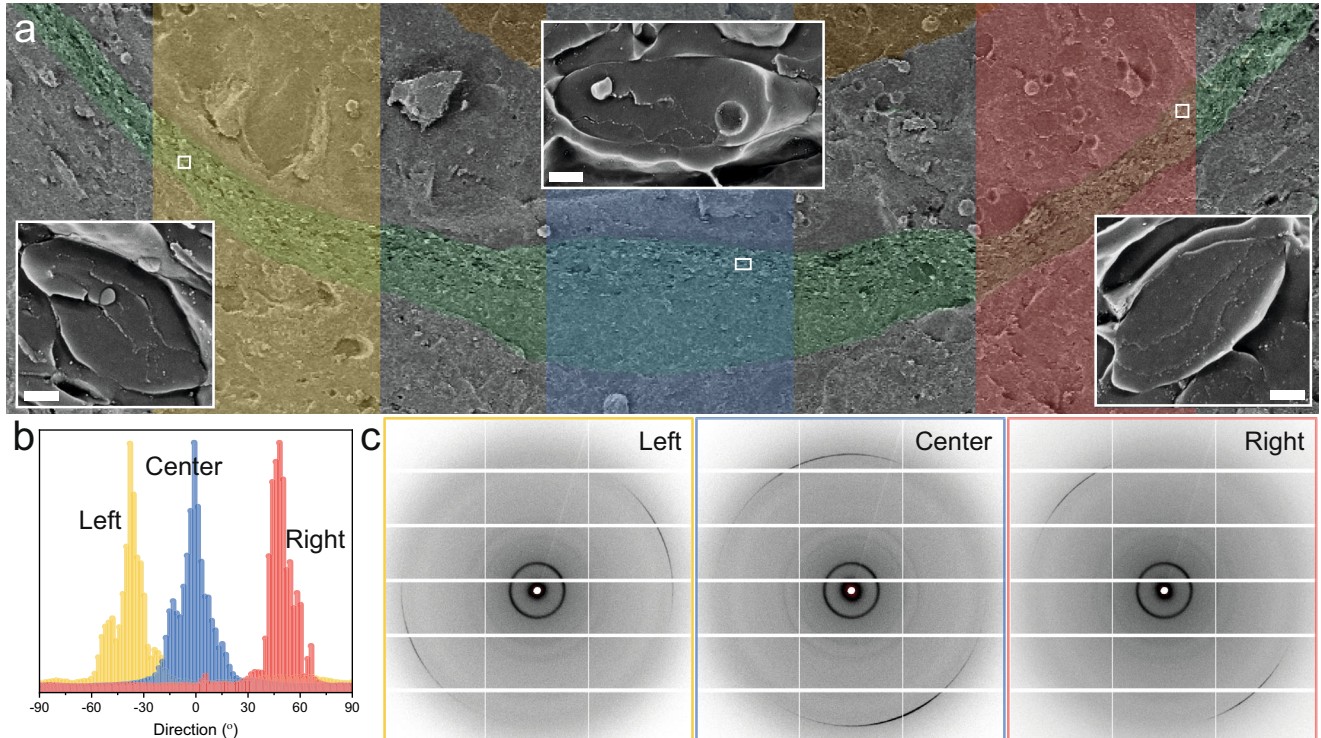

**Fig. 5 | Intraluminal sheets control crystal orientation. a** Low magnification cryo-SEM image of the mirror in a juvenile scallop eye (Fig. 1b). Inserts: representative cryo-SEM images of iridosomes from the regions of interest, demonstrating the orientation of the sheets follows the curvature of the mirror. **b** Histograms representing the directionality of the templating sheets in the juvenile, from the three highlighted regions of interest in (**b**). **c** Representative in situ μspot wide-angle X-ray diffraction (WAXS) patterns from corresponding regions of the adult mirror showing the preferred orientation of the guanine (100) peak across the mirror. Throughout the figure, the colors correspond to the different regions of the mirror: yellow−left, blue−center, and pink−right. Scale bars: 200 nm.

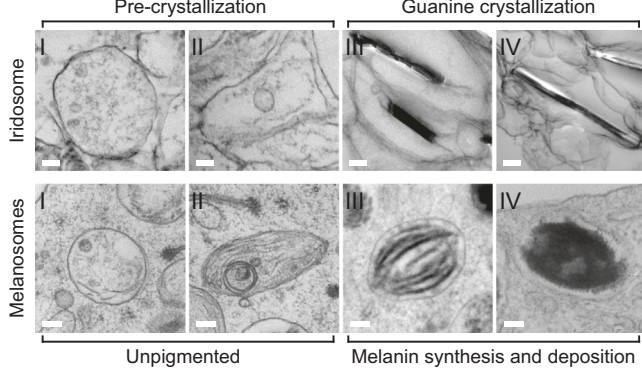

**Fig. 6 | A comparison of iridosome and melanosome organellogenesis.** (Top) TEM images of iridosomes from juvenile scallop eyes: Stage I iridosomes are spherical compartments containing ILVs and disorganized fibrils. Stage II iridosomes elongate concomitantly with the formation of two fibrillar sheets extending across the organelle. In stage III, the sheets template the nucleation and growth of a guanine crystal and become bound to its (100) face. In stage IV, the growing guanine crystal stretches the organelle membrane tightly around itself. (Bottom) TEM images of melanosomes from high-pressure frozen MNT1 highly pigmented melanoma cells: Stage I melanosomes are spherical endosomal compartments containing ILVs which seed the formation of fibrils. These fibrils elongate and assemble into sheets of the PMEL protein, which stretch the organelle into an ellipsoid−stage II. The sheets template melanin deposition in stages III and IV. In stage IV melanosomes, the intercalated fibers are completely obscured by deposited melanin. Scale bars; 100 nm.

orientation and organization. By mapping the orientation of sheets across different regions of the juvenile mirror (Fig. 5a, b), we determined that even prior to nucleation, the sheets closely align with the curvature of the mirror. Thus, the directionality of the sheets in a juvenile scallop eye corresponds to the orientation of the crystals in the adult obtained by in situ synchrotron XRD (Fig. 5c). This indicates that crystal orientation is "pre-programmed" into the early iridosomes. Thus, the sheets are responsible for directing the highly reflective (100) crystal faces to the incident light and maintaining a smooth gradient of crystal orientations across the mirror surface−minimizing optical aberrations from surface defects[14]. Presumably, some "communication" between organelles is required to generate the tessellated suprastructure, which is most probably achieved via the direction of cytoskeletal elements.

## Discussion

By following iridosome organellogenesis in developing scallop eyes we show that pre-assembled macromolecular sheets within iridosomes template crystal nucleation and growth and direct crystal shape and orientation. Following nucleation, the sheets cap the (100) crystal faces, inhibiting growth along the π-stacking direction to generate optically functional (100) plates. In contrast, when grown in vitro from aqueous solutions, guanine crystals grow preferentially along π-stacking direction to form relatively low-refractive index prisms. The extent to which the fibrous sheets control nucleation and 'shape' the crystals cannot be determined unambiguously from static electron micrographs. It is plausible that the crystal and fibers work in 'concert' together (i.e., each reshaping the other) and that there is a subtle interplay of mechanical interactions between the soft fibers and hard crystals that generate the final morphology. This is evidenced by the wavy nature of the sheets prior to nucleation, which become rigid upon binding to the forming crystal.

The exact nature of the nucleation mechanism is also unknown. The sheets delineate a volume within the iridosome in which the guanine crystals emerge, and likely provide an interface on which the planar face of the guanine molecules nucleate. The occasional

observation of multiple nucleation events in a single iridosome suggests that nucleation can occur at any location along the sheets rather than at specific nucleation sites. Multiple nucleation occurs only rarely because once one crystal forms, it likely becomes the energetically preferred site for attachment and growth.

Another fundamental finding is the striking resemblance between the morphogenesis of guanine in scallops (Fig. 6; top row) and the well-documented melanosome organellogenesis in vertebrates (Fig. 6; bottom row)[30-32]. The four stages of melanosome formation in human MNT1 cells are shown alongside the key stages of guanine morphogenesis derived from Fig. 2. Melanosome formation begins with a spherical, endosomal compartment (a pre-melanosome) containing small ILVs that seed fibril formation[27] (Fig. 6; bottom−stage I). These ILVs carry apolipoprotein E (ApoE) at the ILV membrane, which regulates premelanosome protein (PMEL) loading on the ILVs and facilitates nucleation and assembly of PMEL amyloid fibrils[33]. The size and texture of these ILVs and the presence of disorganized fibrils are highly reminiscent of the multi-vesicular early iridosome (Fig. 6; top−stage I, Fig. 2a, and Supplementary Fig. 5). In stage II of melanosome formation (Fig. 6; bottom−stage II), PMEL amyloid sheets assemble and stretch the organelle into an ellipsoid. As well as dictating melanosome shape, PMEL fibrils function to sequester highly reactive oxidative intermediates produced during melanin synthesis[34]. In iridosomes, prior to nucleation, two highly oriented, intraluminal sheets assemble concomitantly with its transformation into an ellipsoid (Fig. 6; top−stage II). The 2D sheets then template the nucleation and growth of guanine (Fig. 6; top−stage III). This is analogous to stage III melanosomes, where tyrosine is oxidized to melanin, which is then deposited on pre-assembled PMEL sheets (Fig. 6; bottom−stage III). In iridosomes, we observe ILVs (type ii and type iii) in stages II and III which, by analogy to melanosome formation[30,31], likely shuttle enzymes related to guanine synthesis into the organelle. Though the chemistry of the iridosome sheets is unknown, their width[27] and the analogies to melanosome PMEL suggest that they may be formed from a functional amyloid protein. Finally, following melanin and guanine deposition, the templating sheets remain an integral part of the organelle; being intercalated inside the pigment in melanosomes (Fig. 6; bottom−stage IV), and being fused to the guanine surface in iridosomes (Fig. 6; top−stage IV, S4).

Due to their crystallinity and unique optical and physical properties, organic bio-crystals were, until now, usually treated within the framework of biomineralization, not pigmentation. However, our results show that guanine crystal formation exhibits many parallels with the morphogenesis of other lysosome-related pigment organelles. Specifically, we show that a common control mechanism likely underlies guanine formation in scallops and melanin formation in vertebrates. This work uncovers a long-sought explanation to how high refractive index, plate-like biogenic guanine crystals are formed. Previous reports on guanine crystallization[17,29,35,36] showed that in some organisms, guanine forms "non-classically" via a disordered precursor phase but the morphology control mechanism remained unknown. The presence of fibrillar sheets, the coalescence of platelets, and the reshaping of the iridosome membrane by the crystal have been observed in other organisms, suggesting that some aspects of guanine bio-crystallization could be universal.

Template-directed crystallization is utilized in many inorganic biominerals to control crystal orientation and properties[37,38]. However, its role in organic bio-crystallization was not known hitherto. Identifying the chemistry of the intraluminal sheets and vesicles and determining how widespread this mechanism is in biology, will be key to the development of the field of organic biomineralization. Furthermore, understanding how macromolecular templates interact with biogenic organic crystals promises to yield new approaches for controlling the morphological properties of molecular materials[39].

## Methods

### Specimen collection and preparation

Juvenile *Pecten maximus* scallops[40] of different sizes were purchased from Scalpro AS. Scalpro AS produce scallop spat at their facilities in Rong, Øygarden County, Norway. Scallops from the local area were conditioned, spawned, and larvae and spat raised to various sizes from settled larvae to 5−20 mm shell height. Spat are reared by Algetun AS and Hotate AS on the west coast of Norway to larger sizes (2−7 cm). With the help of Prof. Thorolf Magnesen from the University of Bergen, mantles were removed and preserved in a fixative solution of 4% paraformaldehyde (PFA) and 2% glutaraldehyde (GA) in 1xPBS. In the case of small scallops, less than 2 cm, the whole animal was fixed.

### Optical microscopy

A Zeiss Discovery.V20 stereomicroscope equipped with an Axiocam 305 color camera was used to take images of whole scallops, mantles, and eyes.

### Cryogenic scanning electron microscopy (cryo-SEM)−sample preparation and imaging

Eyes from fixed scallops were sandwiched between two aluminum disks and cryo-immobilized in a high-pressure freezing (HPF) device (EM ICE, Leica). The frozen samples were then mounted on a holder under liquid nitrogen in a specialized loading station (EM VCM, Leica) and transferred under cryogenic conditions (EM VCT500, Leica) to a sample preparation freeze fracture device (EM ACE900, Leica). These samples are held under vacuum at a temperature of −120 °C. Samples are fractured by hitting the top disc carrier with a tungsten knife at a speed of 160 mm/s, this exposed a clean fracture plane which can be imaged. Then the samples were etched for 5 min at −110 °C to uncover additional structural details of the sample by controlled evaporation of water. Lastly, the samples were coated with 3 nm of PtC. Samples were imaged in an HRSEM Gemini 300 SEM (Zeiss) by secondary electron in-lens detector while maintaining an operating temperature of −120 °C. During this study, a total of 28 scallop eyes were examined by cryo-SEM.

### Preparation of ultra-thin tissue sections for TEM and STEM imaging of iridosomes

Chemically fixed scallop eyes were washed with 0.1 M cacodylate buffer for 2 h and post-fixed with 1% osmium tetroxide and 2% uranyl acetate according to the method published in ref.[17]. Ultra-thin sections were prepared with an ultra-microtome (RMC, Arizona, USA) and imaged with HRSEM Gemini 300 SEM (Zeiss) STEM detector, and Thermo Fisher Scientific (former FEI) Talos F200C transmission electron microscope operating at 200 kV. The images were taken with Ceta 16 M CMOS camera. The electron diffraction patterns were obtained with a Thermo Fisher Scientific (FEI) Tecnai T12 G$^2$ TWIN TEM operating at 120 kV. Images and electron diffraction (ED) patterns were recorded using a Gatan 794 MultiScan CCD camera. ED analysis was done using Gatan Digital Micrograph software with the DIFPack module. Images and ED patterns were recorded with consideration of potential beam damage to the sample, thus appropriate illumination conditions (spot size) were used to avoid it.

### High-pressure freezing, sample preparation, and conventional EM imaging of melanosomes

MNT1, highly pigmented melanoma cells were seeded on carbonated sapphire disks and grown for 2 days. Cells were then immobilized by HPF using either an HPM 100 (Leica Microsystems, Germany) or an HPM Live μ (CryoCapCell, France), and freeze substituted in anhydrous acetone containing 1% $OsO_4$/2% $H_2O$ for 64 h using the Automatic Freeze Substitution unit (AFS; Leica Microsystems). The samples were included in EMbed 812. Ultra-thin sections were contrasted with an aqueous solution of uranyl acetate and Reynold's lead citrate solution.

Electron micrographs were acquired using a Transmission Electron Microscope (Tecnai Spirit; Thermo Fisher, Eindhoven, The Netherlands) operated at 80 kV, and equipped with a 4k CCD camera (Quemesa, EMSIS, Muenster, Germany).

## TEM tomography: acquisition and analysis

TEM imaging and tomography were performed with Thermo Fisher Scientific (former FEI) Talos F200C transmission electron microscope operating at 200 kV. The images were taken with Ceta 16 M CMOS camera. The tilt series were acquired with Thermo Fisher Scientific Tomography software (version 5.9) between angles varying from −60° to 60° with 1° steps. Before each acquisition of an image and before the movement of the stage to the next tilt angle, a cross-correlation of tracking and focusing were made. The post-processing and the stack alignment were done with Thermo Fisher Scientific Inspect3D software (version 4.4). Later this was refined manually with the software Amira version 6.5 (Geometry Transforms/Align Slices). Movies were created on the movie creation panel of the animation director of Amira 6.5 (FEI, USA). Brightness and contrast levels were adjusted, and the movie was performed by animating the xy-ortho slice through the volume. Then it was exported with 25 frames per second, in the highest quality and as a MPEG file. Due to the low contrast between the cell wall, crystal, and filament, it was not possible to perform an automatic segmentation. Therefore, manual segmentation of the vesicle membrane, fibrils, and crystal was performed using IMOD 4.11 software[41,42].

## Determining the long-range orientational ordering of the intraluminal sheets

Cryo-SEM images of different regions of the mirror in a juvenile scallop eye were used for this analysis. The positions and orientations of the sheets were manually marked by representative lines. The resultant images underwent Fast Fourier Transformed (FFT) analysis using the directionality plugin in FIJI ImageJ (https://imagej.net/plugins/directionality)[43]. This plugin is used to infer the preferred orientation of structures in an image, which is visualized by a representative histogram.

## Synchrotron radiation in situ μspot wide-angle X-ray diffraction

Wide-angle XRD (WAXD) measurements (Fig. 5c) were performed at the mySpot beamline, in the BESSY II synchrotron at Helmholtz–Zentrum Berlin für Materialien und Energie. Chemically fixed adult scallop eyes were mounted and sealed between (X-ray transparent) Kapton foil windows with a rectangular aluminum or lead frame. The samples were kept hydrated throughout the measurement with a drop of water inside the Kapton windows. As was described in detail previously[17], an energy of 15 keV ($\lambda = 0.826561$ Å) was selected by a Mo/BC multilayer monochromator. The 2D WAXD patterns were measured using a Dectrix Eiger X 9 M area detector ($3000 \times 3000$ pixel, 75 μm pixel size). A polycrystalline quartz standard was used to calibrate the beam center and the sample-detector distance. Visualization of the 2D scattering patterns was performed using DPDAK (Version 1.4.2). The data were normalized with respect to the primary beam monitor (ionization chamber) and corrected for the background because of the pinhole and air scattering.

## Statistics and reproducibility

In total, in this study, 28 eyes from 15 scallops (2 cm in diameter) were examined by cryo-SEM. Fractures through these eyes reveal thousands of individual vesicles, which were used to deduce the morphogenesis described in this manuscript. Representative images are shown in Figs. 1, 2, 4, 5, 6 and Supplementary Figs. 2, 3, 4, 5, 7, 8. For the ILV size analysis in Supplementary Fig. 5, $n = 60$ ILVs from independent iridosomes. For the orientational analysis in Fig. 5, 200 vesicles were analyzed. In total, six eyes from six scallops were examined by TEM.

## Reporting summary

Further information on research design is available in the Nature Portfolio Reporting Summary linked to this article.

## Data availability

All the data that support the findings of this study are available from the corresponding author upon request.

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

## Acknowledgements
We thank Profs. Steve Weiner, Lia Addadi, and Michael S. Marks for stimulating discussions on this project. We acknowledge Alexey Tachlytski from Zeiss for his instruction regarding STEM operation and Tally Kossovsky from the Hebrew University of Jerusalem for her guidance and training on microtome operation. We acknowledge Lee Shelly for photographing the juvenile scallops (insert Fig. 1a). We acknowledge Dr. Ivo Zizak and the BESSY II synchrotron, Helmholtz–Zentrum Berlin für Materialien und Energie, Germany, for the provision of synchrotron time at the μspot beamline. A.W. is grateful to the Azrieli Foundation for the award of an Azrieli Graduate Fellowship 2022/23. B.A.P. is the Nahum Guzik Presidential Recruit and the recipient of the 2019 Azrieli Faculty Fellowship. This work was supported by an ERC Starting Grant (Grant number: 852948, "CRYSTALEYES") awarded to B.A.P.

## Author contributions
Conceptualization: A.W. and B.A.P. Methodology: A.W., A.U., and B.A.P. Investigation: A.W., A.U., R.M., E.Z., and G.R. Resources: T.M. Visualization: A.W. and R.M. Funding acquisition: B.A.P. Supervision: B.A.P. Writing—original draft: A.W. and B.A.P. Writing—review and editing: all authors.

## Competing interests
The authors declare no competing interests.
