## [Peer Review File · Nature Communications]

Macromolecular Sheets Direct the Morphology and Orientation of Plate-like Biogenic Guanine CrystalsReviewers' Comments:

Reviewer #1:

Remarks to the Author:

This paper reports a detailed investigation on the formation of iridosomes in developing scallop eyes, revealing that the morphogenesis of iridosomes resembles the melanosome morphogenesis in vertebrates. It is elucidated that pre-assembled intravesicular fibrillar sheets template the nucleation, growth, and orientation of guanine crystals thus leading to the formation of thermodynamically disfavored plate-like crystals that show certain optical functions. This is a sound work but the overall novelty seems not high enough considering that there is a similar work on the formation of biogenic plate-like guanine crystals, which was recently reported in bioRxiv-Biophysics (DOI: 10.1101/2022.09.29.510168). This manuscript may be publishable elsewhere after addressing the following issues.

1. Fig.3a is the same as the inset in Fig.1c.
2. The cryo-samples for the cryo-SEM observation look somewhat damaged, as indicated by the holes in Fig. 1f, the cracks in Fig. 1c, and the protrusion of crystals in many figures. Therefore, some related conclusions are not very convincing. For example, "In adult scallops, the iridosome membrane and intraluminal fibrils ultimately become fused to the crystal surface (Fig. 3, Fig. 1C, insert), assuming its faceted shape¹⁴"---- this conclusion seems speculative.
3. It would be helpful for comparison if the directionality of the mirror region and crystals in Fig.5a and c could be marked as in Fig. 5b.
4. Fig. 2b shows the existence of many small particles. Is it possible that these particles are made of guanine?
5. The thickness of guanine crystals in Fig. C-E is almost same. Then, the growth of guanine may be still along the a axis in biominerals, which is consistent with the habit of guanine crystals. If so, the conclusion "the formation of optically functional, but thermodynamically disfavored plate-like habits" may be not exactly true since the growth process may be still a thermodynamically process.
6. The authors may want to present a schematic illustration of the mechanism for the templated crystallization of biogenic guanine crystals.

Reviewer #2:

Remarks to the Author:

The formation of exquisite guanine crystals by organisms is an exciting field of biomaterial study. The contribution by Wagner et al. presents new insights into the formation process of these spectacular crystals in the scallop eye. The authors use electron microscopy at cryogenic conditions to explore the cellular environment of these crystals as they form and mature. The main finding is the observation of fibrillar layers that envelope the crystals inside the membranous compartment that hosts their formation. The authors propose that these organic structures are functional in directing crystal growth and orienting the growing crystals such that they can fulfill their optical function. This notion is very intriguing and relates to several other biominerals that might be regulated by organic scaffolds. The work provides novel observations from a group that is leading the study of biogenic molecular crystals and it may facilitates a new level of mechanistic understanding in the future.

In order to improve the quality of the work I suggest addressing the following points:

1. The samples were chemically fixed and stained prior to the microscopic observations. This necessitates a degree of caution in interpreting structural relationships between the crystals and the fibrillar layers. It is important to make this clear and maybe tone down a bit the conclusions that are presented throughout the paper.
2. In relation to the first point, there is a logical problem of causality when trying to extract a mechanism from still images. It is difficult to differentiate the scenario in which the fibrils are controlling the crystallization from the scenario where crystallization is shaping the fibrils. Also here, it will be better to leave more flexibility in the interpretations.

3. It is clear that the work would benefit greatly from characterizing the biochemical nature of the fibrils. I am wondering if the authors could use their very interesting comparison to melanin formation and try to investigate something about the biochemical nature of these fibrils. In addition, how the proposed mechanism relates to the chemistry inside the organelle? Guanine is hardly soluble so how is it reaching the fibrillar sheets? How this scenario relates to the previous report on the crystallization pathway of guanine?

More specific comments:

1. "Melanosome morphogenesis" appears both in the title and abstract. I appreciate that the analogy to melanin is very informative in the discussion, but since it is not very widely known and not part of the study, I think it will be better to replace this terminology with something more widely understood.
2. Figure 1 A – it is unclear what the caption means by "an adult and juvenile scallop", there is only one animal in the image.
3. in the following paragraph the text uses terminology of "Type i \ ii \ iii" it is unclear what is meant by these labels.
4. During the first parts the text is referring to "fibers" and only at Figure 3 there is a revision of this notion to "fibrillar sheets". It will be helpful to use the right term from the start.

Reviewer #3:

Remarks to the Author:

The formation process of guanine crystals in the eyes of scallops was investigated by using cryo-SEM, TEM, in-situ WAXS in this work. It is found that the pre-assembled introvesicular fibrillary sheets template crystal nucleation, growth and orientation. The finding is interesting and important since the formation mechanism of twinning guanine crystals in biological systems is not very known up to now. A few questions and comments are as follows.

Figure 1 (A) Stereomicroscope images of the upper mantle in an adult and juvenile scallop... is the image in Figure 1 A (not the inset) the mantle of an adult or a juvenile scallop?

Figure 2 caption. I would suggest to add the exact stages for the different growth processes. for example, A-B precrystallization, how large is the sizes of the scallops and the eyes? (By the way, "(A)-(C)" should be changed to "(A)-(B)" since "(C)-(E) are Nucleation and growth"). I would suggest to add the detailed information about the sizes of the eyes and the body of the different stages to the figure caption.

"In adult scallops, the iridosome membrane and intraluminal fibrils ultimately become fused to the crystal surface (Fig. 3, Fig. 1C, insert), assuming its faceted shape¹⁴ " Figure 3 caption "We assign the inner enveloping layer to the fibrillary sheets observed in juvenile scallop iridosomes which become fused to the crystal surface." These sentences describe the assumptions about the relationship of the iridosome membrane and the crystals, but not the final conclusion. It is better to clarify that these are hypotheses but not conclusion.

Figure 4 shows the "nucleation and crystal growth". Please define the "immature guanine crystal" more clearly. And, add the information about the sizes of the eyes and the body at this stage.

Response Letter to Reviewers

We thank the Reviewers for their very comments which have greatly assisted us in improving the manuscript. We have responded here to each of the comments and have highlighted in yellow the altered parts of the manuscript and SI.

Reviewer #1 (Remarks to the Author):

This paper reports a detailed investigation on the formation of iridosomes in developing scallop eyes, revealing that the morphogenesis of iridosomes resembles the melanosome morphogenesis in vertebrates. It is elucidated that pre-assembled intravesicular fibrillar sheets template the nucleation, growth, and orientation of guanine crystals thus leading to the formation of thermodynamically disfavored plate-like crystals that show certain optical functions. This is a sound work but the overall novelty seems not high enough considering that there is a similar work on the formation of biogenic plate-like guanine crystals, which was recently reported in bioRxiv-Biophysics (DOI: 10.1101/2022.09.29.510168). This manuscript may be publishable elsewhere after addressing the following issues.

We thank the reviewer for their assessment of our manuscript. Regarding novelty, then the present manuscript was submitted to Nat. Commun., prior to the bioRxiv article being deposited. Since the bioRxiv work is on an online archive repository and was submitted after our work was under consideration, it does not impact the novelty of our findings. We note that this is also in agreement with the opinion of Nat. Commun. We hope that given this information the reviewer will agree that this paper reports a novel and important discovery.

We are confident that this work points out, for the first time, two key features relating to the regulation of biogenic guanine crystal formation: (i) the presence of pre-formed templating fibrillar sheets as a means of generating and assembling optically useful, plate-like crystals and, (ii) the finding that iridosome morphogenesis closely resembles melanosome organogenesis, which we anticipate will pave the way towards elucidating a deep understanding of the biochemical control mechanisms underpinning guanine biocrystallization.

1. Fig.3a is the same as the inset in Fig.1c.

We have replaced the insert in Fig. 1C with an alternative image. Our intention here is to show that the mature crystals are tightly enveloped by a de-limiting lipid membrane as documented previously by refs. 18, 21 and 25 in the manuscript and refs. 3-8 in the supplementary information. The rest of the paper documents the morphogenesis of the crystals and shows how

this membrane comes to be closely attached to the surface of the crystals during maturation – due to re-shaping of the iridosome membrane by the growing crystal.

2. The cryo-samples for the cryo-SEM observation look somewhat damaged, as indicated by the holes in Fig. 1f, the cracks in Fig. 1c, and the protrusion of crystals in many figures. Therefore, some related conclusions are not very convincing. For example, “In adult scallops, the iridosome membrane and intraluminal fibrils ultimately become fused to the crystal surface (Fig. 3, Fig. 1C, insert), assuming its faceted shape¹⁴”---- this conclusion seems speculative.

Based on our experience with cryo-SEM, we concluded that the tissues are remarkably well-preserved. This is evidenced by the presence of intact intra-vesicular features (e.g., intraluminal vesicles, fibrillar sheets). The presence of these features shows that the tissues are fully hydrated and in a close-to-life state. Furthermore, we note that low-resolution images (Fig. 1B) show that the eye tissue is not distorted in any way, which may occur due to mechanical damage or dehydration effects.

The cryo-SEM technique involves freeze-fracturing a high-pressure-frozen tissue in a dedicated device (EM ACE900, Leica). This fracture occurs under vacuum at -120 °C and is a result of shear forces exerted on the top half of the sample carrier. This technique generates a clean surface through the sample for imaging. The nature of this fracture is somewhat random and thus inherently will lead to some vesicles being fractured, some remaining unfractured and others dislodged. The presence of protruding crystals is not an artefact. Their presence is revealed by the fracture plane, which removes surrounding tissue/water in their immediate vicinity. We obtained many hundreds of images from 28 scallop eyes, providing thousands of examples of vesicles for examination, which show a very consistent picture.

To elaborate on this point, we have added more information to the Materials and Methods about the cryo-SEM technique, as well as an additional Supplementary Figure (Fig. 4) showing other examples of fractured vesicles.

Despite our confidence in the cryo-SEM results, we agree with the referee that any single imaging technique comes with its own limitations. One of the main reasons we performed TEM imaging in parallel to cryo-SEM, was to obtain a comparative set of images using an alternative technique (as can be seen in the two-column figure 2). The features observed in cryo-SEM (e.g., intraluminal vesicles, fibrillar sheets) are also observed in TEM. Indeed, the results from the two techniques are in exceptionally close agreement.

Based on these data and arguments we are confident in the conclusions drawn in the paper. However, we agree that the specific sentence (mentioned also by reviewer 3) needed modification. We have thus revised the text accordingly together with additional references. This revised text now clearly explains which parts of these observations were previously known, which parts are raw results and which parts are hypothesized, as follows:

“In adult scallops, the iridosome membrane (Fig. 3 white arrow) becomes attached to the crystal surface, assuming its faceted shape²⁵. Upon closer examination, it is possible to discern a second, inner layer fused directly to the crystal surface (Fig. 3 red arrow). We presume that this layer is composed of the intraluminal fibrils found in immature iridosomes that, together with the vesicle membrane, form a two-tiered envelope around the crystal. This finding provides a possible explanation to previous reports of ‘double-membrane’ delimited iridosomes in spiders, fish, amphibians, and reptiles¹⁸ (see supplementary references 3-8 and Supplementary Fig. 6).”

3. It would be helpful for comparison if the directionality of the mirror region and crystals in Fig.5a and c could be marked as in Fig. 5b.

We thank the reviewer for their comments and have amended the figure accordingly.

4. Fig. 2b shows the existence of many small particles. Is it possible that these particles are made of guanine?

This is a very interesting possibility that we have considered. At this stage, we do not know what the various intraluminal vesicles contain. However, they are clearly related to the formation of guanine crystals. It is impossible to say confidently at this stage whether they carry guanine, a guanine precursor, or a related enzyme. However, the analogy to melanosomes would suggest that the smallest vesicles (Fig. 2A, ILV type i) are related to fibrillar sheet formation (ref. 27 in manuscript). In the case of melanosomes, the small vesicles carry apolipoprotein E (ApoE) at the ILV membrane, which regulates PMEL loading on the ILVs and facilitates nucleation and assembly of PMEL amyloid fibrils (ref. 33 in manuscript).

In our work, based on their size on location within the organelle, we designate two other larger vesicles (Fig. 2B, S3, ILV types ii, iii). It is likely that these contain enzymes related to guanine synthesis.

We have added text to the results and discussion with additional references on these points.

5. The thickness of guanine crystals in Fig. C-E is almost same. Then, the growth of guanine may be still along the *a* axis in biominerals, which is consistent with the habit of guanine crystals. If so, the conclusion “the formation of optically functional, but thermodynamically disfavored plate-like habits” may be not exactly true since the growth process may be still a thermodynamically process.

We thank the reviewer for this insightful comment. We agree that the crystals reach their near-final thickness at early growth stages. This means that the crystals do grow quickly along the *a* axis (i.e., in the π stacking direction) but are then redirected by the fibrillar sheets. The presence of these sheets then guides the growth along the H-bonded *bc* direction. Our intention was not to comment on whether the growth is a thermodynamic process but rather to point out that the final crystal habit is different from the favored prismatic habit obtained from BFDH calculations (ref. 11 in manuscript) and from re-crystallization in water (Cryst. Growth Des. 2016, 16, 4975–4980; ACS Materials Lett. 2020, 2, 446–452).

We agree that the text was somewhat confusing and required clarification. We have modified the relevant parts of the manuscript accordingly.

6. The authors may want to present a schematic illustration of the mechanism for the templated crystallization of biogenic guanine crystals.

We contemplated adding a schematic to the paper but decided that it was best to allow the raw images to tell the story themselves. However, we have modified Figure 6 with more annotations to make the morphogenetic parallelism clearer.

Reviewer #2 (Remarks to the Author):

The formation of exquisite guanine crystals by organisms is an exciting field of biomaterial study. The contribution by Wagner et al. presents new insights into the formation process of these spectacular crystals in the scallop eye. The authors use electron microscopy at cryogenic conditions to explore the cellular environment of these crystals as they form and mature. The main finding is the observation of fibrillar layers that envelope the crystals inside the membranous compartment that hosts their formation. The authors propose that these organic structures are functional in directing crystal growth and orienting the growing crystals such that they can fulfill their optical function. This notion is very intriguing and relates to several other biominerals that might be regulated by organic scaffolds. The work provides novel observations from a group that is leading the study of biogenic molecular crystals and it may facilitate a new level of mechanistic understanding in the future.

We thank the reviewer the favorable assessment of our work.

In order to improve the quality of the work I suggest addressing the following points:

1. The samples were chemically fixed and stained prior to the microscopic observations. This necessitates a degree of caution in interpreting structural relationships between the crystals and the fibrillar layers. It is important to make this clear and maybe tone down a bit the conclusions that are presented throughout the paper. 2. In relation to the first point, there is a logical problem of causality when trying to extract a mechanism from still images. It is difficult to differentiate the scenario in which the fibrils are controlling the crystallization from the scenario where crystallization is shaping the fibrils. Also here, it will be better to leave more flexibility in the interpretations.

We thank the reviewer for these important comments. We are confident that our approach of using high-pressure-freezing/cryo-SEM on chemically fixed tissues preserves the samples in a close-to-native state. As stated in our answer to reviewer 1, the presence of intravesicular features such as intraluminal vesicles and fibers is evidence of good tissue preservation. Moreover, observing the same phenomena in both TEM and cryo-SEM requiring two different preparation methods adds strength to our conclusions. However, we agree with the reviewer's comments on causality and accept that caution is required when interpreting any microscopic data.

We have reworded the manuscript in several places to address the causality issue.

The observation of pre-assembled and oriented (see Fig. 5) fibrillar sheets *prior* to crystallization, indicates that they act as a template for controlling nucleation, growth and orientation – i.e., crystal orientation is ‘programmed’ into the system before nucleation occurs. However, we agree that the crystal and fibers likely work in ‘concert’ together and that there is a subtle interplay of mechanical interactions that generate the final morphology.

In addition to re-wording the ms in several places, we have added the following statement to the discussion:

“...The extent to which the fibrous sheets physically ‘shape’ the crystals cannot be determined unambiguously from static electron micrographs. It is plausible that the crystal and fibers work in ‘concert’ together (i.e., each re-shaping the other) and that there is a subtle interplay of mechanical interactions between the soft fibers and hard crystals that generates the final morphology...”

3. It is clear that the work would benefit greatly from characterizing the biochemical nature of the fibrils. I am wondering if the authors could use their very interesting comparison to melanin formation and try to investigate something about the biochemical nature of these fibrils. In addition, how the proposed mechanism relates to the chemistry inside the organelle? Guanine is hardly soluble so how is it reaching the fibrillar sheets? How this scenario relates to the previous report on the crystallization pathway of guanine?

We agree that these are incredibly interesting and important questions, and we are pursuing many of them in our laboratory. Nothing is known about the chemistry inside the organelle and how this impacts crystal formation or crystallization pathway. A full chemical characterization of the (i) fibrils, (ii) intraluminal vesicles and (iii) guanine transport mechanisms (addressing the key question of solubility) represent an enormous challenge which we are currently tackling. We believe these questions warrant their own independent reports. The current paper documents for the first time the morphogenesis of guanine organelles, and in so-doing, illuminates the specific questions which now must be pursued by the developing field.

Throughout the results we have made reference to previous reports on guanine crystallization pathway. We have also included an additional paragraph to the discussion as follows:

“...Previous reports on guanine crystallization^{17,29,35,36} showed that in some organisms, guanine forms ‘non-classically’ *via* a disordered precursor phase but the morphology control mechanism remained unknown. The presence of fibrillar sheets, the coalescence of platelets,

and the reshaping of the iridosome membrane by the crystal have been observed in other organisms, suggesting that some aspects of guanine bio-crystallization could be universal...”

More specific comments:

1. “Melanosome morphogenesis” appears both in the title and abstract. I appreciate that the analogy to melanin is very informative in the discussion, but since it is not very widely known and not part of the study, I think it will be better to replace this terminology with something more widely understood.

We agree with this comment and have altered the title of the manuscript to try and make it more accessible to a wider audience. Our proposed modified title is as follows:

‘Macromolecular Sheets Direct the Morphology and Orientation of Plate-like Biogenic Guanine Crystals’

We have also modified the abstract to further clarify these aspects.

2. Figure 1 A – it is unclear what the caption means by “an adult and juvenile scallop”, there is only one animal in the image.

We have modified the figure in accordance with the reviewer’s suggestions.

3. in the following paragraph the text uses terminology of “Type i \ ii \ iii” it is unclear what is meant by these labels.

We thank the reviewer for his comment. During our imaging we observed a few types of intraluminal vesicles (ILVs). We categorized them into three types based on their size, stage of appearance, and interaction with other components in the crystal vesicle. We have expanded and clarified the relevant results section (with additional references), expanded in the discussion and modified Supplementary Fig. 5 to assist with this.

Modified results section:

“...The small ILVs (type *i* ILVs) which have a rough surface texture (Fig. 2A insert and Supplementary Fig. 5) are observed only in spherical iridosomes in the earliest stages of formation. During the next formation stage, the iridosomes elongate concomitantly with the formation of two intraluminal fibrils which stretch from one pole of the vesicle to the other (Fig. 2B)²⁸. Two types of intraluminal vesicles are seen in these iridosomes, distinguished by their size and degree of association with the intraluminal fibrils. Type *ii* ILVs (80-100 nm) are

closely bound to the organized fibrils, and type *iii* ILVs (250-325 nm) are typically found near the iridosome membrane and do not associate closely with the fibrils (Supplementary Fig. 5)..."

4. During the first parts the text is referring to “fibers” and only at Figure 3 there is a revision of this notion to “fibrillar sheets”. It will be helpful to use the right term from the start.

We thank the reviewer for this comment. This was done intentionally because it is only with the 3D tomography results (rather than the 2D imaging results) that we could conclusively prove the sheet-like nature of the fibrils.

Reviewer #3 (Remarks to the Author):

The formation process of guanine crystals in the eyes of scallops was investigated by using cryo-SEM, TEM, in-situ WAXS in this work. It is found that the pre-assembled introvesicular fibrillary sheets template crystal nucleation, growth and orientation. The finding is interesting and important since the formation mechanism of twinning guanine crystals in biological systems is not very known up to now. A few questions and comments are as follows.

We thank the reviewer the favorable assessment of our work.

Figure 1 (A) Stereomicroscope images of the upper mantle in an adult and juvenile scallop... is the image in Figure 1 A (not the inset) the mantle of an adult or a juvenile scallop?

We have modified the Figure 1A in keeping with this comment.

Figure 2 caption. I would suggest to add the exact stages for the different growth processes. for example, A-B precrystallization, how large is the sizes of the scallops and the eyes? (By the way, "(A)-(C)" should be changed to "(A)-(B)" since "(C)-(E) are Nucleation and growth"). I would suggest to add the detailed information about the sizes of the eyes and the body of the different stages to the figure caption.

Thank you for this comment, we have added the required information to the caption. The eyes examined in the study were approximately 200 μm in diameter and they all had forming mirrors. The different stages of crystal growth are observed in a single developmental stage of the organism (i.e., a forming iridophore cell in a juvenile scallop contains thousands of iridosomes at different developmental states). We noticed that more mature iridosomes were found at the distal edge of the cell, whereas immature iridosomes were observed more proximally. This enabled us to derive the morphogenetic sequence by observing many iridosomes in juvenile animals.

We have added information to the figure caption, modified the results, and added an additional supplementary figure to demonstrate this - Supplementary Fig. 3.

“In adult scallops, the iridosome membrane and intraluminal fibrils ultimately become fused to the crystal surface (Fig. 3, Fig. 1C, insert), assuming its faceted shape¹⁴” Figure 3 caption
“We assign the inner enveloping layer to the fibrillary sheets observed in juvenile scallop iridosomes which become fused to the crystal surface.” These sentences describe the

assumptions about the relationship of the iridosome membrane and the crystals, but not the final conclusion. It is better to clarify that these are hypotheses but not conclusion.

We thank you for this comment. We agree that these sentences were not well worded. We have clarified this in the revised manuscript, which addresses similar comments by Reviewer 1.

Figure 4 shows the “nucleation and crystal growth”. Please define the “immature guanine crystal” more clearly. And, add the information about the sizes of the eyes and the body at this stage.

Thank you for this comment. We have made the requires changes to the revised manuscript.

Reviewers' Comments:

Reviewer #1:

Remarks to the Author:

Considering that the deposited bioRxiv work does not impact the novelty of the work reported in this manuscript, I would agree that the findings presented here are overall novel enough for publication in Nat. Commun. I also note that the authors have addressed my concerns in an appropriate way. Therefore, I recommend acceptance of this manuscript for publication.

Reviewer #2:

Remarks to the Author:

I carefully read the revised version of the manuscript. It includes some important, even though minor, clarifications, but does not add any substantially new information. This makes me conclude that at present the evidence cannot support the full assertion of the research. In other words, what is convincingly reported in the work are the finding of the organic scaffolds inside the organelle and the anatomical similarities to melanosome. These are very interesting observations, but they do not support the claims that the control over nucleation and growth originates from these structures. My view doesn't imply that the authors' assertion is incorrect, only that it isn't backed by enough direct evidence. For example, the scaffolds are wavy before crystal nucleation, so how only one crystal forms and not multiple crystals that will line the scaffold? Of course, it is possible that the waviness is a preservation artifact, but this option also stands for the evidence supporting the postulated scenario. On the same lines, how the scaffolds can explain the observation of many platelets? This is not an expected consequence of nucleation from a template that also function as a barrier. Therefore, even though I really like the work and find it interesting, I think the authors need to go further in limiting their statements to evidence-backed conclusions. The notion of crystallization control is currently in the title, abstract, lines 61, 145-6, 170-1, 177-8, which are all part of the Results section. This makes the work too speculative and suggestive of a mechanism when the observations are providing only structural aspects (which are very interesting by themselves).

Reviewer #3:

Remarks to the Author:

The revised manuscript was improved a lot and the authors have addressed all the questions and comments from the reviewers. I would suggest to publish it as it is now.

Response Letter to Reviewers

We thank the Reviewers for their comments which have assisted us in improving the manuscript. We have responded here in blue to each of the comments and have highlighted in yellow the altered parts in the manuscript.

Reviewer #1 (Remarks to the Author):

Considering that the deposited bioRxiv work does not impact the novelty of the work reported in this manuscript, I would agree that the findings presented here are overall novel enough for publication in Nat. Commun. I also note that the authors have addressed my concerns in an appropriate way. Therefore, I recommend acceptance of this manuscript for publication.

We thank the Reviewer for the favorable assessment of our manuscript.

Reviewer #2 (Remarks to the Author):

I carefully read the revised version of the manuscript. It includes some important, even though minor, clarifications, but does not add any substantially new information. This makes me conclude that at present the evidence cannot support the full assertion of the research. In other words, what is convincingly reported in the work are the finding of the organic scaffolds inside the organelle and the anatomical similarities to melanosome. These are very interesting observations, but they do not support the claims that the control over nucleation and growth originates from these structures. My view doesn't imply that the authors' assertion is incorrect, only that it isn't backed by enough direct evidence.

We thank the Reviewer for their very thoughtful comments on this matter. We have answered each concern here and have made corresponding changes highlighted in the manuscript.

- For example, the scaffolds are wavy before crystal nucleation, so how only one crystal forms and not multiple crystals that will line the scaffold? Of course, it is possible that the waviness is a preservation artifact, but this option also stands for the evidence supporting the postulated scenario.

The evidence suggests that the sheets perform three essential functions relating to nucleation and growth: (i) to delineate a volume in which crystal nucleation and growth occurs in the iridosome, (ii) to provide an interface upon which guanine nucleates on the planar face of the molecule, (iii) to cap the (100) faces of the growing crystals following nucleation, preventing crystal growth along the pi-stacking direction. The occasional observation of multiple

nucleation events (Supplementary Fig. 8) in a single iridosome indicates that nucleation is not restricted to specific sites along the scaffolds. The rarity of multiple nucleation is likely because, once a crystal forms it becomes the energetically preferred site for attachment rather than the sheets.

The Reviewer is correct in saying that the wavy nature of the sheets prior to nucleation indicates that they too are re-shaped by the forming crystal and that they do not ‘physically’ mold the crystal into a plate.

We have carefully modified the manuscript in numerous places to address these issues and have added a new section to the discussion as follows:

“The extent to which the fibrous sheets control nucleation and ‘shape’ the crystals cannot be determined unambiguously from static electron micrographs. It is plausible that the crystal and fibers work in ‘concert’ together (i.e., each re-shaping the other) and that there is a subtle interplay of mechanical interactions between the soft fibers and hard crystals that generates the final morphology. This is evidenced by the wavy nature of the sheets prior to nucleation, which become rigid upon binding to the forming crystal.

The exact nature of the nucleation mechanism is also unknown. The sheets delineate a volume within the iridosome in which the guanine crystals emerge, and likely provide an interface on which the planar face of the guanine molecules nucleate. The occasional observation of multiple nucleation events in a single iridosome suggests that nucleation can occur at any location along the sheets rather than at specific nucleation sites. Multiple nucleation occurs only rarely because once one crystal forms it likely becomes the energetically preferred site for attachment and growth.”

- On the same lines, how the scaffolds can explain the observation of many platelets? This is not an expected consequence of nucleation from a template that also function as a barrier.

The Reviewer is correct. The evidence suggests that the platelet nature of the crystals is not a consequence of nucleation from the template but is an intrinsic (uncontrolled) feature of guanine crystallization. We added one sentence in the results section to clarify this in the manuscript:

“... The platelet nature of the crystals may be an intrinsic feature of guanine crystallization occurring independently of the sheets. Similar platelets which merge during crystallization are

also observed in spiders¹⁷ and lizards²⁹, indicating that this is a universal feature of guanine bio-crystallization”

- Therefore, even though I really like the work and find it interesting, I think the authors need to go further in limiting their statements to evidence-backed conclusions. The notion of crystallization control is currently in the title, abstract, lines 61, 145-6, 170-1, 177-8, which are all part of the Results section. This makes the work too speculative and suggestive of a mechanism when the observations are providing only structural aspects (which are very interesting by themselves).

We agree with the referee that ‘causality’ and control need to be treated carefully in these instances. In the previous revision of the manuscript, we were careful to stay away from words such as “control” and “force” to limit our assertions. The use of “template” and “direct” are intended to express the possibility of a more subtle and nuanced interaction between the sheets and the forming crystals.

We have modified the relevant sentences in the manuscript (including deleting several instances of the word ‘control’) to try and better express these points (highlighted in yellow).

Reviewer #3 (Remarks to the Author):

The revised manuscript was improved a lot and the authors have addressed all the questions and comments from the reviewers. I would suggest to publish it as it is now.

We thank the Reviewer for the favorable assessment of our manuscript.